# On the Model Shrinkage Effect of Gamma Process Edge Partition Models

**Iku Ohama**[*‡]    **Issei Sato**[†]    **Takuya Kida**[‡]    **Hiroki Arimura**[‡]
[*]Panasonic Corp., Japan  [†]The Univ. of Tokyo, Japan  [‡]Hokkaido Univ., Japan
ohama.iku@jp.panasonic.com   sato@k.u-tokyo.ac.jp   {kida,arim}@ist.hokudai.ac.jp

## Abstract

The edge partition model (EPM) is a fundamental Bayesian nonparametric model for extracting an overlapping structure from binary matrix. The EPM adopts a gamma process ($\Gamma$P) prior to automatically shrink the number of active atoms. However, we empirically found that the model shrinkage of the EPM does not typically work appropriately and leads to an overfitted solution. An analysis of the expectation of the EPM's intensity function suggested that the gamma priors for the EPM hyperparameters disturb the model shrinkage effect of the internal $\Gamma$P. In order to ensure that the model shrinkage effect of the EPM works in an appropriate manner, we proposed two novel generative constructions of the EPM: CEPM incorporating constrained gamma priors, and DEPM incorporating Dirichlet priors instead of the gamma priors. Furthermore, all DEPM's model parameters including the infinite atoms of the $\Gamma$P prior could be marginalized out, and thus it was possible to derive a truly infinite DEPM (IDEPM) that can be efficiently inferred using a collapsed Gibbs sampler. We experimentally confirmed that the model shrinkage of the proposed models works well and that the IDEPM indicated state-of-the-art performance in generalization ability, link prediction accuracy, mixing efficiency, and convergence speed.

## 1 Introduction

Discovering low-dimensional structure from a binary matrix is an important problem in relational data analysis. Bayesian nonparametric priors, such as Dirichlet process (DP) [1] and hierarchical Dirichlet process (HDP) [2], have been widely applied to construct statistical models with an automatic model shrinkage effect [3, 4]. Recently, more advanced stochastic processes such as the Indian buffet process (IBP) [5] enabled the construction of statistical models for discovering overlapping structures [6, 7], wherein each individual in a data matrix can belong to multiple latent classes.

Among these models, the *edge partition model* (EPM) [8] is a fundamental Bayesian nonparametric model for extracting overlapping latent structure underlying a given binary matrix. The EPM considers latent positive random counts for only non-zero entries in a given binary matrix and factorizes the count matrix into two non-negative matrices and a non-negative diagonal matrix. A link probability of the EPM for an entry is defined by transforming the multiplication of the non-negative matrices into a probability, and thus the EPM can capture overlapping structures with a noisy-OR manner [6]. By incorporating a gamma process ($\Gamma$P) as a prior for the diagonal matrix, the number of active atoms of the EPM shrinks automatically according to the given data. Furthermore, by truncating the infinite atoms of the $\Gamma$P with a finite number, all parameters and hyperparameters of the EPM can be inferred using closed-form Gibbs sampler. Although, the EPM is well designed to capture an overlapping structure and has an attractive affinity with a closed-form posterior

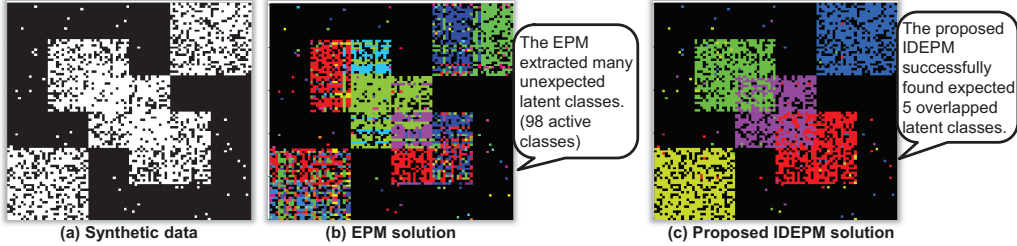

Figure 1: (Best viewed in color.) A synthetic example: (a) synthetic $90 \times 90$ data (white corresponds to one, and black to zero); (b) EPM solution; and (c) the proposed IDEPM solution. In (b) and (c), non-zero entries are colored to indicate their most probable assignment to the latent classes.

inference, the EPM involves a critical drawback in its model shrinkage mechanism. As we experimentally show in Sec. 5, we found that the model shrinkage effect of the EPM does not typically work in an appropriate manner. Figure 1 shows a synthetic example. As shown in Fig. 1a, there are five overlapping latent classes (white blocks). However, as shown in Fig. 1b, the EPM overestimates the number of active atoms (classes) and overfits the data.

In this paper, we analyze the undesired property of the EPM's model shrinkage mechanism and propose novel generative constructions for the EPM to overcome the aforementioned disadvantage. As shown in Fig. 1c, the IDEPM proposed in this paper successfully shrinks unnecessary atoms. More specifically, we have three major contributions in this paper.

(1) We analyse the generative construction of the EPM and find a property that disturbs its model shrinkage effect (Sec. 3). We derive the expectation of the EPM's intensity function (Theorem 1), which is the total sum of the infinite atoms for an entry. From the derived expectation, we obtain a new finding that gamma priors for the EPM's hyperparameters disturb the model shrinkage effect of the internal ΓP (Theorem 2). That is, the derived expectation is expressed by a multiplication of the terms related to ΓP and other gamma priors. Thus, there is no guarantee that the expected number of active atoms is finite.

(2) Based on the analysis of the EPM's intensity function, we propose two novel constructions of the EPM: the CEPM incorporating constrained gamma priors (Sec. 4.1) and the DEPM incorporating Dirichlet priors instead of the gamma priors (Sec. 4.2). The model shrinkage effect of the CEPM and DEPM works appropriately because the expectation of their intensity functions depends only on the ΓP prior (Sec. 4.1 and Theorem 3 in Sec. 4.2).

(3) Furthermore, for the DEPM, all model parameters, including the infinite atoms of the ΓP prior, can be marginalized out (Theorem 4). Therefore, we can derive a truly infinite DEPM (IDEPM), which has a closed-form marginal likelihood without truncating infinite atoms, and can be efficiently inferred using collapsed Gibbs sampler [9] (Sec. 4.3).

## 2   The Edge Partition Model (EPM)

In this section, we review the EPM [8] as a baseline model. Let $\boldsymbol{x}$ be an $I \times J$ binary matrix, where an entry between $i$-th row and $j$-th column is represented by $x_{i,j} \in \{0, 1\}$. In order to extract an overlapping structure underlying $\boldsymbol{x}$, the EPM [8] considers a non-negative matrix factorization problem on latent Poisson counts as follows:

$$x_{i,j} = \mathbb{I}(m_{i,j,\cdot} \geq 1), \quad m_{i,j,\cdot} \,|\, \boldsymbol{U}, \boldsymbol{V}, \boldsymbol{\lambda} \sim \text{Poisson}\left(\sum_{k=1}^{K} U_{i,k} V_{j,k} \lambda_k\right), \qquad (1)$$

where $\boldsymbol{U}$ and $\boldsymbol{V}$ are $I \times K$ and $J \times K$ non-negative matrices, respectively, and $\boldsymbol{\lambda}$ is a $K \times K$ non-negative diagonal matrix. Note that $\mathbb{I}(\cdot)$ is 1 if the predicate holds and is zero otherwise. The latent counts $\boldsymbol{m}$ take positive values only for edges (non-zero entries) within a given binary matrix and the generative model for each positive count is equivalently expressed as a sum of $K$ Poisson random variables as $m_{i,j,\cdot} = \sum_k m_{i,j,k}, m_{i,j,k} \sim \text{Poisson}(U_{i,k} V_{j,k} \lambda_k)$. This is the reason why the above model is called edge partition model. Marginalizing $\boldsymbol{m}$ out from Eq. (1), the generative model of the EPM can be equivalently rewritten as

$x_{i,j} \mid \boldsymbol{U}, \boldsymbol{V}, \boldsymbol{\lambda} \sim \text{Bernoulli}(1 - \prod_k e^{-U_{i,k} V_{j,k} \lambda_k})$. As $e^{-U_{i,k} V_{j,k} \lambda_k} \in [0, 1]$ denotes the probability that a Poisson random variable with mean $U_{i,k} V_{j,k} \lambda_k$ corresponds to zero, the EPM can capture an overlapping structure with a noisy-OR manner [6].

In order to complete the Bayesian hierarchical model of the EPM, gamma priors are adopted as $U_{i,k} \sim \text{Gamma}(a_1, b_1)$ and $V_{j,k} \sim \text{Gamma}(a_2, b_2)$, where $a_1, a_2$ are shape parameters and $b_1, b_2$ are rate parameters for the gamma distribution, respectively. Furthermore, a gamma process (ΓP) is incorporated as a Bayesian nonparametric prior for $\boldsymbol{\lambda}$ to make the EPM automatically shrink its number of atoms $K$. Let $\text{Gamma}(\gamma_0/T, c_0)$ denote a truncated ΓP with a concentration parameter $\gamma_0$ and a rate parameter $c_0$, where $T$ denotes a truncation level that should be set large enough to ensure a good approximation to the true ΓP. Then, the diagonal elements of $\boldsymbol{\lambda}$ are drawn as $\lambda_k \sim \text{Gamma}(\gamma_0/T, c_0)$ for $k \in \{1, \dots, T\}$.

The posterior inference for all parameters and hyperparameters of the EPM can be performed using Gibbs sampler (detailed in Appendix A). Thanks to the conjugacy between gamma and Poisson distributions, given $m_{i,\cdot,k} = \sum_j m_{i,j,k}$ and $m_{\cdot,j,k} = \sum_i m_{i,j,k}$, posterior sampling for $U_{i,k}$ and $V_{j,k}$ is straightforward. As the ΓP prior is approximated by a gamma distribution, posterior sampling for $\lambda_k$ also can be performed straightforwardly. Given $\boldsymbol{U}$, $\boldsymbol{V}$, and $\boldsymbol{\lambda}$, posterior sample for $m_{i,j,\cdot}$ can be simulated using zero-truncated Poisson (ZTP) distribution [10]. Finally, we can obtain sufficient statistics $m_{i,j,k}$ by partitioning $m_{i,j,\cdot}$ into $T$ atoms using a multinomial distribution. Furthermore, all hyperparameters of the EPM (i.e., $\gamma_0$, $c_0$, $a_1$, $a_2$, $b_1$, and $b_2$) can also be sampled by assuming a gamma hyper prior $\text{Gamma}(e_0, f_0)$. Thanks to the conjugacy between gamma distributions, posterior sampling for $c_0$, $b_1$, and $b_2$ is straightforward. For the remaining hyperparameters, we can construct closed-form Gibbs samplers using data augmentation techniques [11, 12, 2].

## 3 Analysis for Model Shrinkage Mechanism

The EPM is well designed to capture an overlapping structure with a simple Gibbs inference. However, the EPM involves a critical drawback in its model shrinkage mechanism.

For the EPM, a ΓP prior is incorporated as a prior for the non-negative diagonal matrix as $\lambda_k \sim \text{Gamma}(\gamma_0/T, c_0)$. From the form of the truncated ΓP, thanks to the additive property of independent gamma random variables, the total sum of $\lambda_k$ over countably infinite atoms follows a gamma distribution as $\sum_{k=1}^{\infty} \lambda_k \sim \text{Gamma}(\gamma_0, c_0)$, wherein the intensity function of the ΓP has a finite expectation as $\mathbb{E}[\sum_{k=1}^{\infty} \lambda_k] = \frac{\gamma_0}{c_0}$. Therefore, the ΓP has a regularization mechanism that automatically shrinks the number of atoms according to given observations.

However, as experimentally shown in Sec. 5, the model shrinkage mechanism of the EPM does not work appropriately. More specifically, the EPM often overestimates the number of active atoms and overfits the data. Thus, we analyse the intensity function of the EPM to reveal the reason why the model shrinkage mechanism does not work appropriately.

**Theorem 1.** *The expectation of the EPM's intensity function $\sum_{k=1}^{\infty} U_{i,k} V_{j,k} \lambda_k$ for an entry $(i, j)$ is finite and can be expressed as follows:*

$$\mathbb{E}\left[\sum_{k=1}^{\infty} U_{i,k} V_{j,k} \lambda_k\right] = \frac{a_1}{b_1} \times \frac{a_2}{b_2} \times \frac{\gamma_0}{c_0}. \tag{2}$$

*Proof.* As $\boldsymbol{U}$, $\boldsymbol{V}$, and $\boldsymbol{\lambda}$ are independent of each other, the expected value operator is multiplicative for the EPM's intensity function. Using the multiplicativity and the low of total expectation, the proof is completed as $\mathbb{E}\left[\sum_{k=1}^{\infty} U_{i,k} V_{j,k} \lambda_k\right] = \sum_{k=1}^{\infty} \mathbb{E}[U_{i,k}]\mathbb{E}[V_{j,k}]\mathbb{E}[\lambda_k] = \frac{a_1}{b_1} \times \frac{a_2}{b_2} \times \mathbb{E}[\sum_{k=1}^{\infty} \lambda_k]$. □

As Eq. (2) in Theorem 1 shows, the expectation of the EPM's intensity function is expressed by multiplying individual expectations of a ΓP and two gamma distributions. This causes an undesirable property to the model shrinkage effect of the EPM. From Theorem 1, another important theorem about the EPM's model shrinkage effect is obtained as follows:

**Theorem 2.** *Given an arbitrary non-negative constant $C$, even if the expectation of the EPM's intensity function in Eq. (2) is fixed as $\mathbb{E}\left[\sum_{k=1}^{\infty} U_{i,k} V_{j,k} \lambda_k\right] = C$, there exist cases in which the model shrinkage effect of the $\Gamma P$ prior disappears.*

*Proof.* Substituting $\mathbb{E}\left[\sum_{k=1}^{\infty} U_{i,k} V_{j,k} \lambda_k\right] = C$ for Eq. (2), we obtain $C = \frac{a_1}{b_1} \times \frac{a_2}{b_2} \times \frac{\gamma_0}{c_0}$. Since $a_1$, $a_2$, $b_1$, and $b_2$ are gamma random variables, even if the expectation of the EPM's intensity function, $C$, is fixed, $\frac{\gamma_0}{c_0}$ can take an arbitrary value so that equation $C = \frac{a_1}{b_1} \times \frac{a_2}{b_2} \times \frac{\gamma_0}{c_0}$ holds. Hence, $\gamma_0$ can take an arbitrary large value such that $\gamma_0 = T \times \widehat{\gamma}_0$. This implies that the $\Gamma P$ prior for the EPM degrades to a gamma distribution without model shrinkage effect as $\lambda_k \sim \mathrm{Gamma}(\gamma_0/T, c_0) = \mathrm{Gamma}(\widehat{\gamma}_0, c_0)$. $\qquad\square$

Theorem 2 indicates that the EPM might overestimate the number of active atoms, and lead to overfitted solutions.

# 4 Proposed Generative Constructions

We describe our novel generative constructions for the EPM with an appropriate model shrinkage effect. According to the analysis described in Sec. 3, the model shrinkage mechanism of the EPM does not work because the expectation of the EPM's intensity function has an undesirable redundancy. This finding motivates the proposal of new generative constructions, in which the expectation of the intensity function depends only on the $\Gamma P$ prior.

First, we propose a naive extension of the original EPM using constrained gamma priors (termed as CEPM). Next, we propose an another generative construction for the EPM by incorporating Dirichlet priors instead of gamma priors (termed as DEPM). Furthermore, for the DEPM, we derive truly infinite DEPM (termed as IDEPM) by marginalizing out all model parameters including the infinite atoms of the $\Gamma P$ prior.

## 4.1 CEPM

In order to ensure that the EPM's intensity function depends solely on the $\Gamma P$ prior, a naive way is to introduce constraints for the hyperparameters of the gamma prior. In the CEPM, the rate parameters of the gamma priors are constrained as $b_1 = C_1 \times a_1$ and $b_2 = C_2 \times a_2$, respectively, where $C_1 > 0$ and $C_2 > 0$ are arbitrary constants. Based on the aforementioned constraints and Theorem 1, the expectation of the intensity function for the CEPM depends only on the $\Gamma P$ prior as $\mathbb{E}[\sum_{k=1}^{\infty} U_{i,k} V_{j,k} \lambda_k] = \frac{\gamma_0}{C_1 C_2 c_0}$.

The posterior inference for the CEPM can be performed using Gibbs sampler in a manner similar to that for the EPM. However, we can not derive closed-form samplers only for $a_1$ and $a_2$ because of the constraints. Thus, in this paper, posterior sampling for $a_1$ and $a_2$ are performed using grid Gibbs sampling [13] (see Appendix B for details).

## 4.2 DEPM

We have another strategy to construct the EPM with efficient model shrinkage effect by re-parametrizing the factorization problem. Let us denote transpose of a matrix $\boldsymbol{A}$ by $\boldsymbol{A}^\top$. According to the generative model of the EPM in Eq. (1), the original generative process for counts $\boldsymbol{m}$ can be viewed as a matrix factorization as $\boldsymbol{m} \approx \boldsymbol{U\lambda V}^\top$. It is clear that the optimal solution of the factorization problem is not unique. Let $\boldsymbol{\Lambda}_1$ and $\boldsymbol{\Lambda}_2$ be arbitrary $K \times K$ non-negative diagonal matrices. If a solution $\boldsymbol{m} \approx \boldsymbol{U\lambda V}^\top$ is globally optimal, then another solution $\boldsymbol{m} \approx (\boldsymbol{U\Lambda}_1)(\boldsymbol{\Lambda}_1^{-1}\boldsymbol{\lambda\Lambda}_2)(\boldsymbol{V\Lambda}_2^{-1})^\top$ is also optimal. In order to ensure that the EPM has only one optimal solution, we re-parametrize the original factorization problem to an equivalent constrained factorization problem as follows:

$$\boldsymbol{m} \approx \boldsymbol{\phi\lambda\psi}^\top, \tag{3}$$

where $\boldsymbol{\phi}$ denotes an $I \times K$ non-negative matrix with $l_1$-constraints as $\sum_i \phi_{i,k} = 1, \forall k$. Similarly, $\boldsymbol{\psi}$ denotes an $J \times K$ non-negative matrix with $l_1$-constraints as $\sum_j \psi_{j,k} = 1, \forall k$. This parameterization ensures the uniqueness of the optimal solution for a given $\boldsymbol{m}$ because each column of $\boldsymbol{\phi}$ and $\boldsymbol{\psi}$ is constrained such that it is defined on a simplex.

According to the factorization in Eq. (3), by incorporating Dirichlet priors instead of gamma priors, the generative construction for $\boldsymbol{m}$ of the DEPM is as follows:

$$m_{i,j,\cdot} \mid \boldsymbol{\phi}, \boldsymbol{\psi}, \boldsymbol{\lambda} \sim \text{Poisson}\left(\sum_{k=1}^{T} \phi_{i,k}\psi_{j,k}\lambda_k\right), \quad \{\phi_{i,k}\}_{i=1}^{I} \mid \alpha_1 \sim \text{Dirichlet}(\overbrace{\alpha_1, \ldots, \alpha_1}^{I}),$$

$$\{\psi_{j,k}\}_{j=1}^{J} \mid \alpha_2 \sim \text{Dirichlet}(\overbrace{\alpha_2, \ldots, \alpha_2}^{J}), \quad \lambda_k \mid \gamma_0, c_0 \sim \text{Gamma}(\gamma_0/T, c_0). \qquad (4)$$

**Theorem 3.** *The expectation of DEPM's intensity function $\sum_{k=1}^{\infty} \phi_{i,k}\psi_{j,k}\lambda_k$ depends sorely on the $\Gamma P$ prior and can be expressed as $\mathbb{E}[\sum_{k=1}^{\infty} \phi_{i,k}\psi_{j,k}\lambda_k] = \frac{\gamma_0}{IJc_0}$.*

*Proof.* The expectations of Dirichlet random variables $\phi_{i,k}$ and $\psi_{j,k}$ are $\frac{1}{I}$ and $\frac{1}{J}$, respectively. Similar to the proof for Theorem 1, using the multiplicativity of independent random variables and the low of total expectation, the proof is completed as $\mathbb{E}\left[\sum_{k=1}^{\infty} \phi_{i,k}\psi_{j,k}\lambda_k\right] = \sum_{k=1}^{\infty} \mathbb{E}[\phi_{i,k}]\mathbb{E}[\psi_{j,k}]\mathbb{E}[\lambda_k] = \frac{1}{I} \times \frac{1}{J} \times \mathbb{E}[\sum_{k=1}^{\infty} \lambda_k].$ $\qquad\square$

Note that, if we set constants $C_1 = I$ and $C_2 = J$ for the CEPM in Sec. 4.1, then the expectation of the intensity function for the CEPM is equivalent to that for the DEPM in Theorem 3. Thus, in order to ensure the fairness of comparisons, we set $C_1 = I$ and $C_2 = J$ for the CEPM in the experiments.

As the Gibbs sampler for $\boldsymbol{\phi}$ and $\boldsymbol{\psi}$ can be derived straightforwardly, the posterior inference for all parameters and hyperparameters of the DEPM also can be performed via closed-form Gibbs sampler (detailed in Appendix C). Differ from the CEPM, $l_1$-constraints in the DEPM ensure the uniqueness of its optimal solution. Thus, the inference for the DEPM is considered as more efficient than that for the CEPM.

### 4.3 Truly Infinite DEPM (IDEPM)

One remarkable property of the DEPM is that we can derive a fully marginalized likelihood function. Similar to the beta-negative binomial topic model [13], we consider a joint distribution for $m_{i,j,\cdot}$ Poisson customers and their assignments $\boldsymbol{z}_{i,j} = \{z_{i,j,s}\}_{s=1}^{m_{i,j,\cdot}} \in \{1, \cdots, T\}^{m_{i,j,\cdot}}$ to $T$ tables as $P(m_{i,j,\cdot}, \boldsymbol{z}_{i,j} \mid \boldsymbol{\phi}, \boldsymbol{\psi}, \boldsymbol{\lambda}) = P(m_{i,j,\cdot} \mid \boldsymbol{\phi}, \boldsymbol{\psi}, \boldsymbol{\lambda}) \prod_{s=1}^{m_{i,j,\cdot}} P(z_{i,j,s} \mid m_{i,j,\cdot}, \boldsymbol{\phi}, \boldsymbol{\psi}, \boldsymbol{\lambda})$. Thanks to the $l_1$-constraints we introduced in Eq. (3), the joint distribution $P(\boldsymbol{m}, \boldsymbol{z} \mid \boldsymbol{\phi}, \boldsymbol{\psi}, \boldsymbol{\lambda})$ has a fully factorized form (see Lemma 1 in Appendix D). Therefore, marginalizing $\boldsymbol{\phi}$, $\boldsymbol{\psi}$, and $\boldsymbol{\lambda}$ out according to the prior construction in Eq. (4), we obtain an analytical marginal likelihood $P(\boldsymbol{m}, \boldsymbol{z})$ for the truncated DEPM (see Appendix D for a detailed derivation).

Furthermore, by taking $T \to \infty$, we can derive a closed-form marginal likelihood for the truly infinite version of the DEPM (termed as IDEPM). In a manner similar to that in [14], we consider the likelihood function for partition $[\boldsymbol{z}]$ instead of the assignments $\boldsymbol{z}$. Assume we have $K_+$ of $T$ atoms for which $m_{\cdot,\cdot,k} = \sum_i \sum_j m_{i,j,k} > 0$, and a partition of $M(= \sum_i \sum_j m_{i,j,\cdot})$ customers into $K_+$ subsets. Then, joint marginal likelihood of the IDEPM for $[\boldsymbol{z}]$ and $\boldsymbol{m}$ is given by the following theorem, with the proof provided in Appendix D:

**Theorem 4.** *The marginal likelihood function of the IDEPM is defined as $P(\boldsymbol{m}, [\boldsymbol{z}])_\infty = \lim_{T\to\infty} P(\boldsymbol{m}, [\boldsymbol{z}]) = \lim_{T\to\infty} \frac{T!}{(T-K_+)!}P(\boldsymbol{m}, \boldsymbol{z})$, and can be derived as follows:*

$$P(\boldsymbol{m}, [\boldsymbol{z}])_\infty = \prod_{i=1}^{I}\prod_{j=1}^{J}\frac{1}{m_{i,j,\cdot}!} \times \prod_{k=1}^{K_+}\frac{\Gamma(I\alpha_1)}{\Gamma(I\alpha_1 + m_{\cdot,\cdot,k})}\prod_{i=1}^{I}\frac{\Gamma(\alpha_1 + m_{i,\cdot,k})}{\Gamma(\alpha_1)}$$

$$\times \prod_{k=1}^{K_+}\frac{\Gamma(J\alpha_2)}{\Gamma(J\alpha_2 + m_{\cdot,\cdot,k})}\prod_{j=1}^{J}\frac{\Gamma(\alpha_2 + m_{\cdot,j,k})}{\Gamma(\alpha_2)} \times \gamma_0^{K_+}\left(\frac{c_0}{c_0+1}\right)^{\gamma_0}\prod_{k=1}^{K_+}\frac{\Gamma(m_{\cdot,\cdot,k})}{(c_0+1)^{m_{\cdot,\cdot,k}}}, \quad (5)$$

*where $m_{i,\cdot,k} = \sum_j m_{i,j,k}$, $m_{\cdot,j,k} = \sum_i m_{i,j,k}$, and $m_{\cdot,\cdot,k} = \sum_i \sum_j m_{i,j,k}$. Note that $\Gamma(\cdot)$ denotes gamma function.*

From Eq. (5) in Theorem 4, we can derive collapsed Gibbs sampler [9] to perform posterior inference for the IDEPM. Since $\boldsymbol{\phi}$, $\boldsymbol{\psi}$, and $\boldsymbol{\lambda}$ have been marginalized out, the only latent variables we have to update are $\boldsymbol{m}$ and $\boldsymbol{z}$.

**Sampling $z$:** Given $m$, similar to the Chinese restaurant process (CRP) [15], the posterior probability that $z_{i,j,s}$ is assigned to $k^*$ is given as follows:

$$P(z_{i,j,s} = k^* \mid \boldsymbol{z}_{\backslash(ijs)}, \boldsymbol{m}) \propto \begin{cases} m_{k^*}^{\backslash(ijs)} \times \frac{\alpha_1 + m_{i,\cdot,k^*}^{\backslash(ijs)}}{I\alpha_1 + m_{\cdot,\cdot,k^*}^{\backslash(ijs)}} \times \frac{\alpha_2 + m_{\cdot,j,k^*}^{\backslash(ijs)}}{I\alpha_2 + m_{\cdot,\cdot,k^*}^{\backslash(ijs)}} & \text{if } m_{\cdot,\cdot,k^*}^{\backslash(ijs)} > 0, \\ \gamma_0 \times \frac{1}{I} \times \frac{1}{J} & \text{if } m_{\cdot,\cdot,k^*}^{\backslash(ijs)} = 0, \end{cases} \quad (6)$$

where the superscript $\backslash(ijs)$ denotes that the corresponding statistics are computed excluding the $s$-th customer of entry $(i, j)$.

**Sampling $m$:** Given $\boldsymbol{z}$, posteriors for the $\boldsymbol{\phi}$ and $\boldsymbol{\psi}$ are simulated as $\{\phi_{i,k}\}_{i=1}^{I} \mid - \sim$ Dirichlet($\{\alpha_1 + m_{i,\cdot,k}\}_{i=1}^{I}$) and $\{\psi_{j,k}\}_{j=1}^{J} \mid - \sim$ Dirichlet($\{\alpha_2 + m_{\cdot,j,k}\}_{j=1}^{J}$) for $k \in \{1, \ldots, K_+\}$. Furthermore, the posterior sampling of the $\lambda_k$ for $K_+$ active atoms can be performed as $\lambda_k \mid - \sim$ Gamma($m_{\cdot,\cdot,k}, c_0 + 1$). Therefore, similar to the sampler for the EPM [8], we can update $\boldsymbol{m}$ as follows:

$$m_{i,j,\cdot} \mid \boldsymbol{\phi}, \boldsymbol{\psi}, \boldsymbol{\lambda} \sim \begin{cases} \delta(0) & \text{if } x_{i,j} = 0, \\ \text{ZTP}(\sum_{k=1}^{K_+} \phi_{i,k} \psi_{j,k} \lambda_k) & \text{if } x_{i,j} = 1, \end{cases} \quad (7)$$

$$\{m_{i,j,k}\}_{k=1}^{K_+} \mid m_{i,j,\cdot}, \boldsymbol{\phi}, \boldsymbol{\psi}, \boldsymbol{\lambda} \sim \text{Multinomial}\left( m_{i,j,\cdot}; \left\{ \frac{\phi_{i,k} \psi_{j,k} \lambda_k}{\sum_{k'=1}^{K_+} \phi_{i,k'} \psi_{j,k'} \lambda_{k'}} \right\}_{k=1}^{K_+} \right), \quad (8)$$

where $\delta(0)$ denotes point mass at zero.

**Sampling hyperparameters:** We can construct closed-form Gibbs sampler for all hyperparameters of the IDEPM assuming a gamma prior (Gamma($e_0, f_0$)). Using the additive property of the $\Gamma$P, posterior sample for the sum of $\lambda_k$ over unused atoms is obtained as $\lambda_{\gamma_0} = \sum_{k'=K_++1}^{\infty} \lambda_{k'} \mid - \sim$ Gamma($\gamma_0, c_0 + 1$). Consequently, we obtain a closed-form posterior sampler for the rate parameter $c_0$ of the $\Gamma$P as $c_0 \mid - \sim$ Gamma($e_0 + \gamma_0, f_0 + \lambda_{\gamma_0} + \sum_{k=1}^{K_+} \lambda_k$). For all remaining hyperparameters (i.e., $\alpha_1$, $\alpha_2$, and $\gamma_0$), we can derive posterior samplers from Eq. (5) using *data augmentation* techniques [12, 8, 2, 11] (detailed in Appendix E).

## 5 Experimental Results

In previous sections, we theoretically analysed the reason why the model shrinkage of the EPM does not work appropriately (Sec. 3) and proposed several novel constructions (i.e., CEPM, DEPM, and IDEPM) of the EPM with an efficient model shrinkage effect (Sec. 4).

The purpose of the experiments involves ascertaining the following hypotheses:

(H1) The original EPM overestimates the number of active atoms and overfits the data. In contrast, the model shrinkage mechanisms of the CEPM and DEPM work appropriately. Consequently, the CEPM and DEPM outperform the EPM in generalization ability and link prediction accuracy.

(H2) Compared with the CEPM, the DEPM indicates better generalization ability and link prediction accuracy because of the uniqueness of the DEPM's optimal solution.

(H3) The IDEPM with collapsed Gibbs sampler is superior to the DEPM in generalization ability, link prediction accuracy, mixing efficiency, and convergence speed.

**Datasets:** The first dataset was the Enron [16] dataset, which comprises e-mails sent between 149 Enron employees. We extracted e-mail transactions from September 2001 and constructed Enron09 dataset. For this dataset, $x_{i,j} = 1(0)$ was used to indicate whether an e-mail was, or was not, sent by the $i$-th employee to the $j$-th employee. For larger dataset, we used the MovieLens [17] dataset, which comprises five-point scale ratings of movies submitted by users. For this dataset, we set $x_{i,j} = 1$ when the rating was higher than three and $x_{i,j} = 0$ otherwise. We prepared two different sized MovieLens dataset: MovieLens100K (943 users and 1,682 movies) and MovieLens1M (6,040 users and 3,706 movies). The densities of the Enron09, MovieLens100K and MovieLens1M datasets were 0.016, 0.035, and 0.026, respectively.

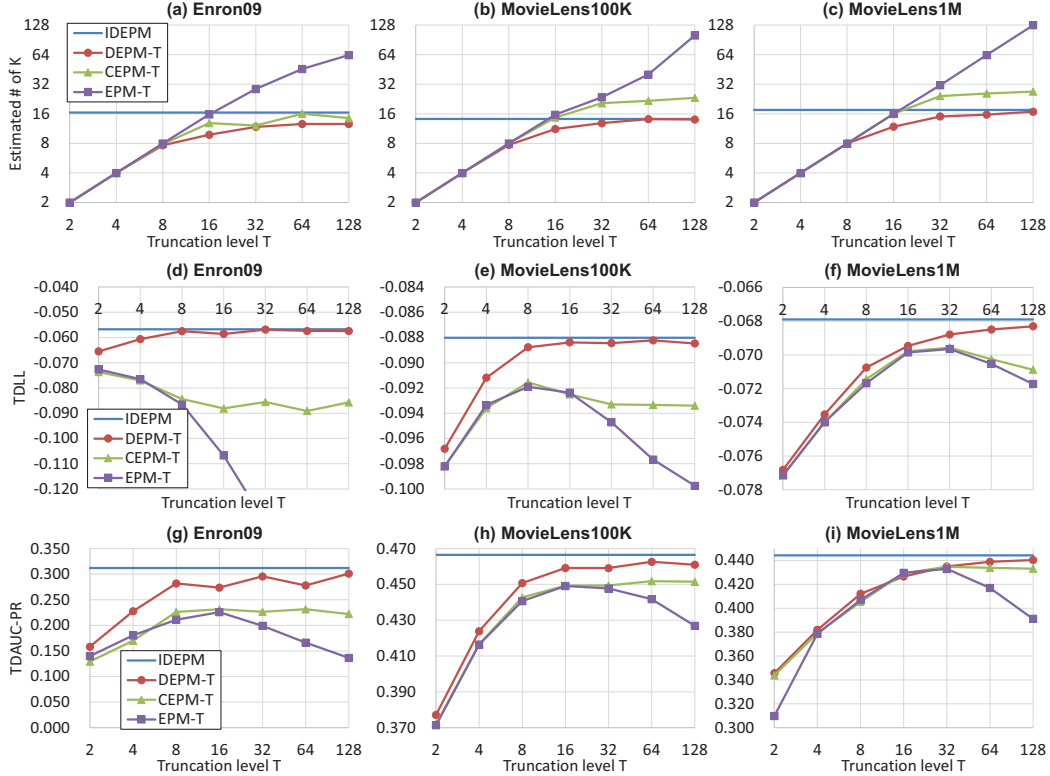

Figure 2: Calculated measurements as functions of the truncation level $T$ for each dataset. The horizontal line in each figure denotes the result obtained using the IDEPM.

**Evaluating Measures:** We adopted three measurements to evaluate the performance of the models. The first is the *estimated number of active atoms $K$* for evaluating the model shrinkage effect of each model. The second is the averaged *Test Data Log Likelihood* (TDLL) for evaluating the generalization ability of each model. We calculated the averaged likelihood that a test entry takes the actual value. For the third measurement, as many real-world binary matrices are often sparse, we adopted the *Test Data Area Under the Curve of the Precision-Recall curve* (TDAUC-PR) [18] to evaluate the link prediction ability. In order to calculate the TDLL and TDAUC-PR, we set all the selected test entries as zero during the inference period, because binary observations for unobserved entries are not observed as missing values but are observed as zeros in many real-world situations.

**Experimental Settings:** Posterior inference for the truncated models (i.e., EPM, CEPM, and DEPM) were performed using standard (non-collapsed) Gibbs sampler. Posterior inference for the IDEPM was performed using the collapsed Gibbs sampler derived in Sec. 4.3. For all models, we also sampled all hyperparameters assuming the same gamma prior (Gamma($e_0, f_0$)). For the purpose of fair comparison, we set hyper-hyperparameters as $e_0 = f_0 = 0.01$ throughout the experiments. We ran 600 Gibbs iterations for each model on each dataset and used the final 100 iterations to calculate the measurements. Furthermore, all reported measurements were averaged values obtained by 10-fold cross validation.

**Results:** Hereafter, the truncated models are denoted as EPM-$T$, CEPM-$T$, and DEPM-$T$ to specify the truncation level $T$. Figure 2 shows the calculated measurements.

(H1) As shown in Figs. 2a–c, the EPM overestimated the number of active atoms $K$ for all datasets especially for a large truncation level $T$. In contrast, the number of active atoms $K$ for the CEPM-$T$ and DEPM-$T$ monotonically converges to a specific value. This result supports the analysis with respect to the relationship between the model shrinkage effect and the expectation of the EPM's intensity function, as discussed in Sec. 3. Consequently,

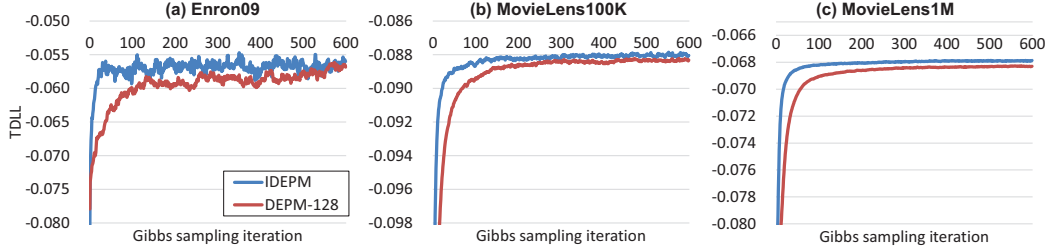

Figure 3: (Best viewed in color.) The TDLL as a function of the Gibbs iterations.

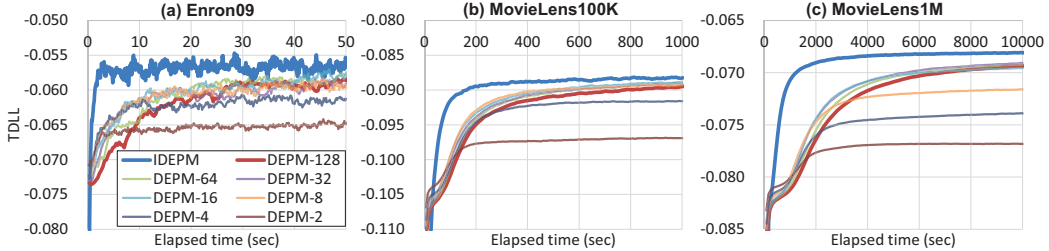

Figure 4: (Best viewed in color.) The TDLL as a function of the elapsed time (in seconds).

as shown by the TDLL (Figs. 2d–f) and TDAUC-PR (Figs. 2g–i), the CEPM and DEPM outperformed the original EPM in both generalization ability and link prediction accuracy.

(H2) As shown in Figs. 2a–c, the model shrinkage effect of the DEPM is stronger than that of the CEPM. As a result, the DEPM significantly outperformed the CEPM in both generalization ability and link prediction accuracy (Figs. 2d–i). Although the CEPM slightly outperformed the EPM, the CEPM with a larger $T$ tends to overfit the data. In contrast, the DEPM indicated its best performance with the largest truncation level ($T = 128$). Therefore, we confirmed that the uniqueness of the optimal solution in the DEPM was considerably important in achieving good generalization ability and link prediction accuracy.

(H3) As shown by the horizontal lines in Figs. 2d–i, the IDEPM indicated the state-of-the-art scores for all datasets. Finally, the computational efficiency of the IDEPM was compared with that of the truncated DEPM. Figure 3 shows the TDLL as a function of the number of Gibbs iterations. In keeping with expectations, the IDEPM indicated significantly better mixing property when compared with that of the DEPM for all datasets. Furthermore, Fig. 4 shows a comparison of the convergence speed of the IDEPM and DEPM with several truncation levels ($T = \{2, 4, 8, 16, 32, 64, 128\}$). As clearly shown in the figure, the convergence of the IDEPM was significantly faster than that of the DEPM with all truncation levels. Therefore, we confirmed that the IDEPM indicated a state-of-the-art performance in generalization ability, link prediction accuracy, mixing efficiency, and convergence speed.

## 6   Conclusions

In this paper, we analysed the model shrinkage effect of the EPM, which is a Bayesian nonparametric model for extracting overlapping structure with an optimal dimension from binary matrices. We derived the expectation of the intensity function of the EPM, and showed that the redundancy of the EPM's intensity function disturbs its model shrinkage effect. According to this finding, we proposed two novel generative construction for the EPM (i.e., CEPM and DEPM) to ensure that its model shrinkage effect works appropriately. Furthermore, we derived a truly infinite version of the DEPM (i.e, IDEPM), which can be inferred using collapsed Gibbs sampler without any approximation for the ΓP. We experimentally showed that the model shrinkage mechanism of the CEPM and DEPM worked appropriately. Furthermore, we confirmed that the proposed IDEPM indicated a state-of-the-art performance in generalization ability, link prediction accuracy, mixing efficiency, and convergence speed. It is of interest to further investigate whether the truly infinite construction of the IDEPM can be applied to more complex and modern machine learning models, including deep brief networks [19], and tensor factorization models [20].

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
