[Supplementary Material]

# Appendix: On the Model Shrinkage Effect of Gamma Process Edge Partition Models

**Iku Ohama**\*‡    **Issei Sato**†    **Takuya Kida**‡    **Hiroki Arimura**‡

\*Panasonic Corp., Japan   †The Univ. of Tokyo, Japan   ‡Hokkaido Univ., Japan

ohama.iku@jp.panasonic.com   sato@k.u-tokyo.ac.jp   {kida,arim}@ist.hokudai.ac.jp

## A  Gibbs Samplers for the EPM

### A.1  Model Description

The full description of the generative model for the EPM [1] is described as follows:

$$x_{i,j} = \mathbb{I}(m_{i,j,\cdot} \geq 1), \quad m_{i,j,\cdot} \,|\, \boldsymbol{U}, \boldsymbol{V}, \boldsymbol{\lambda} \sim \text{Poisson}\left(\sum_{k=1}^{K} U_{i,k} V_{j,k} \lambda_k\right),$$

$$U_{i,k} \sim \text{Gamma}(a_1, b_1), \quad V_{j,k} \sim \text{Gamma}(a_2, b_2), \quad \lambda_k \sim \text{Gamma}(\gamma_0/T, c_0). \qquad (9)$$

### A.2  Closed-form Gibbs Samplers

Posterior inference for all parameters and hyperparameters of the EPM can be performed using Gibbs sampler.

**Sampling $\boldsymbol{m}$:**   From Eq. (9), as $m_{i,j,\cdot} = 0$ if and only if $x_{i,j} = 0$, posterior sampling of $\boldsymbol{m}$ is required only for non-zero entries ($x_{i,j} = 1$), and can be performed using zero-truncated Poisson (ZTP) distribution [2] as follows:

$$m_{i,j,\cdot} \,|\, \boldsymbol{U}, \boldsymbol{\lambda}, \boldsymbol{V} \sim \begin{cases} \delta(0) & \text{if } x_{i,j} = 0, \\ \text{ZTP}(\sum_{k=1}^{T} U_{i,k} \lambda_k V_{j,k}) & \text{if } x_{i,j} = 1. \end{cases} \qquad (10)$$

Then, latent count $m_{i,j,k}$ related to the $k$-th atom can be obtained by partitioning $m_{i,j,\cdot}$ into $T$ atoms as

$$\{m_{i,j,k}\}_{k=1}^{T} \,|\, m_{i,j,\cdot}, \boldsymbol{U}, \boldsymbol{\lambda}, \boldsymbol{V} \sim \text{Multinomial}\left(m_{i,j,\cdot}; \left\{\frac{U_{i,k} \lambda_k V_{j,k}}{\sum_{k'=1}^{T} U_{i,k'} \lambda_{k'} V_{j,k'}}\right\}_{k=1}^{T}\right). \qquad (11)$$

**Sampling $\boldsymbol{U}, \boldsymbol{V}, \boldsymbol{\lambda}$:**   As the generative model for $m_{i,j,k}$ can be given as $m_{i,j,k} \,|\, \boldsymbol{U}, \boldsymbol{V}, \boldsymbol{\lambda} \sim$ Poisson($U_{i,k} V_{j,k} \lambda_k$), according to the additive property of the Poisson distributions, generative models for aggregated counts also can be expressed as follows:

$$m_{i,\cdot,k} = (\textstyle\sum_j m_{i,j,k}) \,|\, \boldsymbol{U}, \boldsymbol{V}, \boldsymbol{\lambda} \sim \text{Poisson}(U_{i,k}(\textstyle\sum_j V_{j,k})\lambda_k), \qquad (12)$$

$$m_{\cdot,j,k} = (\textstyle\sum_i m_{i,j,k}) \,|\, \boldsymbol{U}, \boldsymbol{V}, \boldsymbol{\lambda} \sim \text{Poisson}((\textstyle\sum_i U_{i,k})V_{j,k}\lambda_k), \qquad (13)$$

$$m_{\cdot,\cdot,k} = (\textstyle\sum_i \sum_j m_{i,j,k}) \,|\, \boldsymbol{U}, \boldsymbol{V}, \boldsymbol{\lambda} \sim \text{Poisson}((\textstyle\sum_i U_{i,k})(\textstyle\sum_j V_{j,k})\lambda_k). \qquad (14)$$

Therefore, thanks to the conjugacy between Poisson and gamma distributions, posterior samplers for $\boldsymbol{U}$, $\boldsymbol{V}$, and $\boldsymbol{\lambda}$ are straightforwardly derived as follows:

$$U_{i,k} \,|\, - \sim \text{Gamma}(a_1 + m_{i,\cdot,k}, b_1 + (\textstyle\sum_j V_{j,k})\lambda_k), \qquad (15)$$

$$V_{j,k} \,|\, - \sim \text{Gamma}(a_2 + m_{\cdot,j,k}, b_2 + (\textstyle\sum_i U_{i,k})\lambda_k), \qquad (16)$$

$$\lambda_k \,|\, - \sim \text{Gamma}(\gamma_0/T + m_{\cdot,\cdot,k}, c_0 + (\textstyle\sum_i U_{i,k})(\textstyle\sum_j V_{j,k})). \qquad (17)$$

### A.3 Sampling Hyperparameters

**Sampling $b_1, b_2, c_0$:** Thanks to the conjugacy between gamma distributions, posterior samplers for $b_1$, $b_2$, and $c_0$ are straightforwardly performed as follows:

$$b_1 \mid - \sim \text{Gamma}(e_0 + ITa_1, f_0 + \sum_i \sum_k \phi_{i,k}), \tag{18}$$

$$b_2 \mid - \sim \text{Gamma}(e_0 + JTa_2, f_0 + \sum_j \sum_k \psi_{j,k}), \tag{19}$$

$$c_0 \mid - \sim \text{Gamma}(e_0 + \gamma_0, f_0 + \sum_k \lambda_k). \tag{20}$$

For the remaining hyperparameters (i.e., $a_1$, $a_2$, and $\gamma_0$), we can construct closed-form Gibbs samplers using data augmentation techniques [3, 1, 4, 5], that consider an expanded probability over target and some auxiliary variables. The key strategy is the use of the following expansions:

$$\frac{\Gamma(u)}{\Gamma(u+n)} = \frac{B(u,n)}{\Gamma(n)} = \Gamma(n)^{-1} \int_0^1 v^{u-1}(1-v)^{n-1} dv, \tag{21}$$

$$\frac{\Gamma(u+n)}{\Gamma(u)} = \sum_{w=0}^n S(n,w)u^w, \tag{22}$$

where $B(\cdot, \cdot)$ is the beta function and $S(\cdot, \cdot)$ is the Stirling number of the first kind.

**Sampling $a_1, a_2$:** For shape parameter $a_1$, marginalizing $\boldsymbol{U}$ from Eq. (12), we have a partially marginalized likelihood related to target variable $a_1$ as:

$$P(\{m_{i,\cdot,k}\}_{i,k} \mid \boldsymbol{V}, \boldsymbol{\lambda}) \propto \prod_{k=1}^T \left\{ \left( \frac{b_1}{b_1 + (\sum_j V_{j,k})\lambda_k} \right)^{Ia_1} \prod_{i=1}^I \frac{\Gamma(a_1 + m_{i,\cdot,k})}{\Gamma(a_1)} \right\}. \tag{23}$$

Therefore, expanding Eq. (23) using Eq. (22) and assuming gamma prior as $a_1 \sim \text{Gamma}(e_0, f_0)$, posterior sampling for $a_1$ can be performed as follows:

$$w_{i,k} \mid - \sim \text{Antoniak}(m_{i,\cdot,k}, a_1), \tag{24}$$

$$a_1 \mid - \sim \text{Gamma}\left(e_0 + \sum_i \sum_k w_{i,k}, f_0 - I \times \sum_k \ln \frac{b_1}{b_1 + (\sum_j V_{j,k})}\right), \tag{25}$$

where $\text{Antoniak}(m_{i,\cdot,k}, a_1)$ is an Antoniak distribution [6]. This is the distribution of the number of occupied tables if $m_{i,\cdot,k}$ customers are assigned to one of an infinite number of tables using the Chinese restaurant process (CRP) [7, 8] with concentration parameter $a_1$, and is sampled as $w_{i,k} = \sum_{p=1}^{m_{i,\cdot,k}} w_{i,k,p}, w_{i,k,p} \sim \text{Bernoulli}\left(\frac{a_1}{a_1+p-1}\right)$. Similarly, posterior sampler for $a_2$ can be derived from Eqs. (13) and (22) (omitted for brevity).

**Sampling $\gamma_0$:** Similar to the samplers for $a_1$ and $a_2$, according to Eqs. (14) and (22), $\gamma_0$ can be updated as follows:

$$w_k \mid - \sim \text{Antoniak}(m_{\cdot,\cdot,k}, \gamma_0/T), \tag{26}$$

$$\gamma_0 \mid - \sim \text{Gamma}\left(e_0 + \sum_k w_k, f_0 - \frac{1}{T}\sum_k \ln \frac{c_0}{c_0 + (\sum_i U_{i,k})(\sum_j V_{j,k})}\right). \tag{27}$$

## B Gibbs Samplers for the CEPM

Posterior inference for the CEPM can be performed using Gibbs sampler as same as that for the EPM. However, only $a_1$ and $a_2$ do not have closed-form sampler because of introduced constraints $b_1 = C_1 \times a_1$ and $b_2 = C_2 \times a_2$. Therefore, instead of sampling from true posterior, we use the grid Gibbs sampler [9] to sample from a discrete probability distribution

$$P(a_1 \mid -) \propto \text{Eq (23)} \times P(a_1) \tag{28}$$

over a grid of points $\frac{1}{1+a_1} = 0.01, 0.02, \ldots, 0.99$. Note that $a_2$ can be sampled in a same way as $a_1$ (omitted for brevity).

## C  Gibbs Samplers for the DEPM

### C.1  Closed-form Gibbs Samplers

**Sampling $\phi, \psi$:**  Given $m_{\cdot,\cdot,k} = \sum_i \sum_j m_{i,j,k}$, generative process for latent count $m_{i,\cdot,k}$ can be expressed as

$$\{m_{i,\cdot,k}\}_{i=1}^I \,|\, m_{\cdot,\cdot,k}, \phi, \psi, \lambda \sim \text{Multinomial}\left(m_{\cdot,\cdot,k}; \{\phi_{i,k}\}_{i=1}^I\right). \tag{29}$$

Thanks to conjugacy between Eq. (29) and Dirichlet prior in Eq. (4), posterior sampling for $\phi$ can be performed as

$$\{\phi_{i,k}\}_{i=1}^I \,|\, - \sim \text{Dirichlet}(\{\alpha_1 + m_{i,\cdot,k}\}_{i=1}^I). \tag{30}$$

Similarly, $\psi$ can be updated as

$$\{\psi_{j,k}\}_{j=1}^J \,|\, - \sim \text{Dirichlet}(\{\alpha_2 + m_{\cdot,j,k}\}_{j=1}^J). \tag{31}$$

**Sampling $m, \lambda$:**  Posterior samplers for remaining latent variables $m$ and $\lambda$ are straightforwardly given from Eqs. (10), (11), and (17) by replacing $U$ and $V$ with $\phi$ and $\psi$, respectively.

### C.2  Sampling Hyperparameters

**Sampling $\alpha_1, \alpha_2$:**  Similar to Appendix A.3, marginalizing $\phi$ out from Eq. (4) and expanding the marginal likelihood using Eqs. (21) and (22), posterior sampling for $\alpha_1$ can be derived as follows:

$$v_{1,k} \,|\, - \sim \text{Beta}(I\alpha_1, m_{\cdot,\cdot,k}), \tag{32}$$

$$w_{1,i,k} \,|\, - \sim \text{Antoniak}(m_{i,\cdot,k}, \alpha_1), \tag{33}$$

$$\alpha_1 \,|\, - \sim \text{Gamma}(e_0 + \textstyle\sum_i\sum_k w_{1,i,k}, f_0 - I \times \sum_k \ln v_{1,k}). \tag{34}$$

Note that the posterior sampler for $\alpha_2$ can be derived in same way (omitted for brevity).

**Sampling $\gamma_0, c_0$:**  The remaining hyperparameters (i.e., $\gamma_0$ and $c_0$) can be updated as same as in the EPM. Similar to the sampler for the EPM, $c_0$ can be updated using Eq. (20). Finally, posterior sampler for $\gamma_0$ can be derived as

$$w_k \,|\, - \sim \text{Antoniak}(m_{\cdot,\cdot,k}, \gamma_0/T), \tag{35}$$

$$\gamma_0 \,|\, - \sim \text{Gamma}\left(e_0 + \textstyle\sum_k w_k, f_0 - \ln \frac{c_0}{c_0 + 1}\right). \tag{36}$$

## D  Proof of Theorem 4

Considering a joint distribution for $m_{i,j,\cdot}$ customers and their assignments $z_{i,j} = \{z_{i,j,s}\}_{s=1}^{m_{i,j,\cdot}} \in \{1, \cdots, T\}^{m_{i,j,\cdot}}$ to $T$ tables, we have following lemma for the truncated DEPM:

**Lemma 1.** *The joint distribution over $m$ and $z$ for the DEPM is expressed by a fully factorized form as*

$$P(m, z \,|\, \phi, \psi, \lambda) = \prod_{i=1}^I \prod_{j=1}^J \frac{1}{m_{i,j,\cdot}!} \times \prod_{i=1}^I \prod_{k=1}^T \phi_{i,k}^{m_{i,\cdot,k}} \times \prod_{j=1}^J \prod_{k=1}^T \psi_{j,k}^{m_{\cdot,j,k}} \times \prod_{k=1}^T \lambda_k^{m_{\cdot,\cdot,k}} e^{-\lambda_k}. \tag{37}$$

*Proof.* As the likelihood functions $P(m_{i,j,\cdot} \,|\, \phi, \psi, \lambda)$ and $P(z_{i,j,s} \,|\, m_{i,j,\cdot}, \phi, \psi, \lambda)$ are given as

$$P(m_{i,j,\cdot} \,|\, \phi, \psi, \lambda) = \frac{1}{m_{i,j,\cdot}!} \left(\sum_{k=1}^T \phi_{i,k}\psi_{j,k}\lambda_k\right)^{m_{i,j,\cdot}} e^{-\sum_{k=1}^T \phi_{i,k}\psi_{j,k}\lambda_k}, \tag{38}$$

$$P(z_{i,j,s} = k^* \,|\, m_{i,j,\cdot}, \phi, \psi, \lambda) = \frac{\phi_{i,k^*}\psi_{j,k^*}\lambda_{k^*}}{\sum_{k'=1}^T \phi_{i,k'}\psi_{j,k'}\lambda_{k'}}, \tag{39}$$

respectively, we obtain the joint likelihood function for $\boldsymbol{m}$ and $\boldsymbol{z}$ as follows:

$$P(\boldsymbol{m},\boldsymbol{z}\,|\,\boldsymbol{\phi},\boldsymbol{\psi},\boldsymbol{\lambda})$$
$$=\prod_{i=1}^{I}\prod_{j=1}^{J}\left\{P(m_{i,j,\cdot}\,|\,\boldsymbol{\phi},\boldsymbol{\psi},\boldsymbol{\lambda})\prod_{s=1}^{m_{i,j,\cdot}}P(z_{i,j,s}\,|\,m_{i,j,\cdot},\boldsymbol{\phi},\boldsymbol{\psi},\boldsymbol{\lambda})\right\}$$
$$=\prod_{i=1}^{I}\prod_{j=1}^{J}\frac{1}{m_{i,j,\cdot}!}\times\prod_{i=1}^{I}\prod_{k=1}^{T}\phi_{i,k}^{m_{i,\cdot,k}}\times\prod_{j=1}^{J}\prod_{k=1}^{T}\psi_{j,k}^{m_{\cdot,j,k}}\times\prod_{k=1}^{T}\lambda_{k}^{m_{\cdot,\cdot,k}}e^{-\lambda_{k}(\sum_{i}\phi_{i,k})(\sum_{j}\psi_{j,k})}.$$
(40)

Thanks to the $l_1$-constraints for $\boldsymbol{\phi}$ and $\boldsymbol{\psi}$ we introduced in Eq. (3), substituting $\sum_{i}\phi_{i,k}=\sum_{j}\psi_{j,k}=1$ for Eq. (40), we obtain Eq. (37) in Lemma 1. □

Thanks to the conjugacy between Eq. (37) in Lemma 1 and prior construction in Eq. (4), marginalizing $\boldsymbol{\phi}$, $\boldsymbol{\psi}$, and $\boldsymbol{\lambda}$ out, we obtain the following marginal likelihood for the DEPM:

$$P(\boldsymbol{m},\boldsymbol{z})=\prod_{i=1}^{I}\prod_{j=1}^{J}\frac{1}{m_{i,j,\cdot}!}\times\prod_{k=1}^{T}\frac{\Gamma(I\alpha_1)}{\Gamma(I\alpha_1+m_{\cdot,\cdot,k})}\prod_{i=1}^{I}\frac{\Gamma(\alpha_1+m_{i,\cdot,k})}{\Gamma(\alpha_1)}$$
$$\times\prod_{k=1}^{T}\frac{\Gamma(J\alpha_2)}{\Gamma(J\alpha_2+m_{\cdot,\cdot,k})}\prod_{j=1}^{J}\frac{\Gamma(\alpha_2+m_{\cdot,j,k})}{\Gamma(\alpha_2)}\times\prod_{k=1}^{T}\frac{\Gamma\left(\frac{\gamma_0}{T}+m_{\cdot,\cdot,k}\right)c_0^{\frac{\gamma_0}{T}}}{\Gamma\left(\frac{\gamma_0}{T}\right)(c_0+1)^{\frac{\gamma_0}{T}+m_{\cdot,\cdot,k}}}.\quad(41)$$

Considering a partition $[\boldsymbol{z}]$ instead of the assignments $\boldsymbol{z}$ as same as in [10], the marginal likelihood function $P(\boldsymbol{m},[\boldsymbol{z}])$ for a partition of the truncated DEPM can be expressed as

$$P(\boldsymbol{m},[\boldsymbol{z}])=\frac{T!}{(T-K_+)!}P(\boldsymbol{m},\boldsymbol{z})$$
$$=\prod_{i=1}^{I}\prod_{j=1}^{J}\frac{1}{m_{i,j,\cdot}!}\times\prod_{k=1}^{K_+}\frac{\Gamma(I\alpha_1)}{\Gamma(I\alpha_1+m_{\cdot,\cdot,k})}\prod_{i=1}^{I}\frac{\Gamma(\alpha_1+m_{i,\cdot,k})}{\Gamma(\alpha_1)}$$
$$\times\prod_{k=1}^{K_+}\frac{\Gamma(J\alpha_2)}{\Gamma(J\alpha_2+m_{\cdot,\cdot,k})}\prod_{j=1}^{J}\frac{\Gamma(\alpha_2+m_{\cdot,j,k})}{\Gamma(\alpha_2)}$$
$$\times\frac{T!}{(T-K_+)!T^{K_+}}\times\gamma_0^{K_+}\left(\frac{c_0}{c_0+1}\right)^{\gamma_0}\prod_{k=1}^{K_+}\frac{\prod_{l=1}^{m_{\cdot,\cdot,k}-1}(l+\gamma_0/T)}{(c_0+1)^{m_{\cdot,\cdot,k}}}.\quad(42)$$

Therefore, taking $T\to\infty$ in Eq. (42), we obtain the marginal likelihood function for the truly infinite DEPM (i.e., IDEPM) as in Eq. (5) of Theorem 4.

## E    Sampling Hyperparameters for the IDEPM

**Sampling $\alpha_1,\alpha_2$:**   Posterior samplers for $\alpha_1$ and $\alpha_2$ of the IDEPM are equivalent to those of the truncated DEPM as in Appendix C.2.

**Sampling $\gamma_0$:**   From Eq. (5), we straightforwardly obtain the posterior sampler for $\gamma_0$ as

$$\gamma_0\,|-\sim\text{Gamma}\left(e_0+K_+,f_0-\ln\frac{c_0}{c_0+1}\right).\quad(43)$$

Note that $\gamma_0$ in Eq. (5) can be marginalized out assuming gamma prior. However, we explicitly sample $\gamma_0$ for simplicity in this paper.

**Sampling $c_0$:**   As derived in Sec. 4.3 of main article, $c_0$ is updated as

$$\lambda_k\,|-\sim\text{Gamma}(m_{\cdot,\cdot,k},c_0+1)\quad k\in\{1,\ldots,K_+\},\quad(44)$$
$$\lambda_{\gamma_0}\,|-\sim\text{Gamma}(\gamma_0,c_0+1),\quad(45)$$
$$c_0\,|-\sim\text{Gamma}(e_0+\gamma_0,f_0+\lambda_{\gamma_0}+\textstyle\sum_{k=1}^{K_+}\lambda_k).\quad(46)$$