[Reviews · NeurIPS 2017]

Reviewer 1



This paper presents several variants of GP-EPM to solve the shrinkage problem of GP-EPM for modelling relational data. The proposed models are demonstrated to have better link prediction performance than GP-EPM by estimating the number of latent communities better. Note I always point out this problem to my students as: the model is "non-identifiable". Don't get me wrong, the whole body of work, EPM, etc., is fabulous, but identifiability was always an important lesson in class. The paper is well-written and easy to follow. The proposed models are well-motivated empirically with synthetic examples and then with theoretical analysis. To me, the main contribution of the paper is it points out that the obvious unidentifiability issue in GP-EPM can, interestingly, be a problem in some real cases. The proposed CEPM is just a naive model, while DEPM is a simple but intuitive solution and DEPM to IDEPM is like mapping the Dirichlet distribution to Dirichlet process. The solutions are not novel but straightforward to the problem. Finally the experimental results support the main claims, which makes it an interesting paper. Note, in 4.3 (line 187), you say "one remarkable property". This *should* be well known in the non-parametric community. The result is perhaps the simplest case of the main theorem in "Poisson Latent Feature Calculus for Generalized Indian Buffet Processes", Lancelot James, 2014. The terms in $K_+$ in (5) are the marginal for the gamma process. The theorem can be pretty much written down without derivation using standard results. This is a good paper cleaning up the obvious flaws in EPM and its implementation. It covers good though routine theory, and experimental work. Section 4.3 should be rewritten. However, I belief the important issue of identifiability should be better discussed and more experiments done. For instance, a reviewer points out that the problem is partially overcome more recently using stronger priors. Clearly, identifiability is something we require to make theory easier and priors more interpretable. It is not strictly necessary, though it is routinely expected in the statistics community. Anyway, would be good to see more discussion of the issues and further experimental investigation.

Reviewer 2



This paper introduces refinements to edge partition model (EPM), which is a nonparametric Bayesian method for extracting latent overlapping features from binary relational data. The authors use constraints over node-feature association distributions in order to improve the shrinkage effect in the EPM. The following concerns need to be addressed in my opinion: I am not completely convinced about the argument made in section 3 about the source of poor shrinkage in EPM, where the main reason has been stated as arbitrary values for hyperparameters of feature distributions. Can we simply overcome this by assuming narrow priors? On the other hand, for CEPM, the constants $C_1$ and $C_2$ have been set equal to dimensions for fair comparison. However, I assume the amount of shrinkage will depend on their choice in practice. How do you optimally set them? Regarding DEPM and its infinite version, I think it's a good idea to use Dirichlet distributions for features. However, it's not clear for me how much of performance improvement of DEPM compared to EPM in link prediction is due to shrinkage improvement and not resolving unidentifiability problem using Dirichlet distributions. Finally, while it is clear based on the experiments that EPM has the least shrinkage among the methods, it would be nice if this problem was illustrated on a simulated dataset, were the real number of latent factors is known.

Reviewer 3



The paper proposes some refinement to the infinite edge partition model (EPM) that may suffer from relatively poor shrinkage effect for estimating the number of network components. The paper is an easy read, the observations about this particular limitation of EPM are correct and the experiments are well designed. However, the paper has the following limitations which must be addressed: 1. Please use different notations for "a." and "b." in the sentence on line 80 and for "m_{i,k}", "m_{j,k}" on line 203. 2. The observations about the shrinkage are correct in the paper. However, by imposing priors on the parameters of the gamma distribution, some of these undesired behaviors can be avoided. For example, please review the following papers and the experimental setup therein: -- http://jmlr.org/papers/v17/15-633.html -- https://link.springer.com/chapter/10.1007%2F978-3-319-23528-8_18 -- https://mingyuanzhou.github.io/Papers/DLDA_TLASGR_v12.pdf Perhaps, a more detailed experimental analysis would reveal that the problem with shrinkage is not that critical and can be mitigated. 3. Moreover, several papers have already used the Dirichlet priors to get around this specific problem of shrinkage and identifiability. For example, please review the following paper: -- https://www.cse.iitk.ac.in/users/piyush/papers/topic_kg_aistats.pdf 4. The infinite DEP model is novel. However, the derivations in Section 4.3 are related to the usage of Dirichlet-multinomial distribution. So while the introduction of the collapsed sampler is novel, the derivation of the same may not be considered that novel. 5. Please fix the typo "an another important theorem..".